# Ion Channels as New Attractive Targets to Improve Re-Myelination Processes in the Brain

**DOI:** 10.3390/ijms22147277

**Published:** 2021-07-06

**Authors:** Federica Cherchi, Irene Bulli, Martina Venturini, Anna Maria Pugliese, Elisabetta Coppi

**Affiliations:** Department of Neuroscience, Psychology, Drug Research and Child Health (NEUROFARBA), Section of Pharmacology and Toxicology, University of Florence, 50139 Florence, Italy; federica.cherchi@unifi.it (F.C.); irene.bulli@unifi.it (I.B.); martina.venturini@unifi.it (M.V.); annamaria.pugliese@unifi.it (A.M.P.)

**Keywords:** oligodendrocyte precursor cells, oligodendrocyte differentiation, voltage-gated ion channels, neurotransmitter receptors, purines, glutamate, GABA, myelination, multiple sclerosis, experimental autoimmune encephalomyelitis

## Abstract

Multiple sclerosis (MS) is the most demyelinating disease of the central nervous system (CNS) characterized by neuroinflammation. Oligodendrocyte progenitor cells (OPCs) are cycling cells in the developing and adult CNS that, under demyelinating conditions, migrate to the site of lesions and differentiate into mature oligodendrocytes to remyelinate damaged axons. However, this process fails during disease chronicization due to impaired OPC differentiation. Moreover, OPCs are crucial players in neuro-glial communication as they receive synaptic inputs from neurons and express ion channels and neurotransmitter/neuromodulator receptors that control their maturation. Ion channels are recognized as attractive therapeutic targets, and indeed ligand-gated and voltage-gated channels can both be found among the top five pharmaceutical target groups of FDA-approved agents. Their modulation ameliorates some of the symptoms of MS and improves the outcome of related animal models. However, the exact mechanism of action of ion-channel targeting compounds is often still unclear due to the wide expression of these channels on neurons, glia, and infiltrating immune cells. The present review summarizes recent findings in the field to get further insights into physio-pathophysiological processes and possible therapeutic mechanisms of drug actions.

## 1. Introduction

Oligodendrocytes (OLs) are ramified glial cells within the central nervous system (CNS) whose terminal processes generate myelin and enwrap neuronal axons. Myelin is crucial for the saltatory propagation of electrical impulses along nerve fibres, enabling rapid communication between networks in the CNS [1]. In addition to myelin deposition, OLs secrete metabolic factors and maintain energy homeostasis to support axonal integrity and promote neuronal survival [2]. 

Thus, a deeper understanding of OL functions during brain development, as well as during their regeneration in neurological disorders that involve OL and myelin loss, is crucial to understand their homeostatic functions within the CNS function and to identify new therapeutic targets for demyelinating diseases.

During CNS development, OL progenitor cells (OPCs) are generated from neural stem/progenitor cells (NSPCs) in several regions in a precise spatiotemporal manner [3,4]. Multiple transcriptional regulators cooperate to orchestrate changes in gene expression leading to OPC fate selection and subsequent differentiation into OL. Details on intrinsic signals regulating oligodendroglial cell specification and progression have been described [5,6]. However, since OLs are part of a complex environment containing neurons, astrocytes, microglia, and vascular/perivascular cells, the control of oligodendrogliogenesis likely relies on multiple extrinsic cues and cell–cell interactions during development or regeneration.

## 2. Oligodendrogliogenesis

Only 5–8% of total glial cells are OPC, which are evenly distributed in white and grey matter [7] with different functions; OPCs in white matter present a higher proliferative response to platelet-derived growth factor (PDGF)-A and enhanced differentiation into myelinating OLs than OPC in grey matter [8]. 

Oligodendrogliogenesis and remyelination are crucial events for white matter reorganization [9]. Among other functions, OLs support and regulate axonal electrical activity by producing myelin not only to increase action potential (AP) conduction, but also to provide metabolic support and stabilize axonal cytoskeleton [10,11]. After acute traumatic injury in rodents, there is an active proliferation of OPCs derived from NSPCs in the subventricular zone [12]. Migration of oligodendroglial cells from the proliferative zones to their final position is an essential step during CNS development and myelination. In humans, increased OPC pool at sites of ischemic brain insults has been observed post-mortem. However, only a fraction of proliferating OPCs become mature functional OLs and contribute to remyelination. Furthermore, there is an age-related decline in normal oligodendrogliogenesis. Interestingly, a recent study demonstrated the possibility of exercise to increase OPCs in white matter as a potent therapy for neural circuit remodelling [13]. 

In order to study oligodendrogliogenesis, several markers are available to identify different steps of OPC maturation (Figure 1). Once committed to the oligodendroglial lineage, cell surface antigens can be recognized by specific antibodies, such as A2B5 [14]. Moreover, SRY-Box Transcription Factor 9 (Sox9) is also strongly expressed first in NSPCs, and later in glial cells of the CNS, and is essential for proper development of both OLs and astrocytes; Sox9 continues to be expressed after specification into OPCs [15].

Among the best known OPC markers are PDGFR-α, the receptor for PDGF-A, and the neuron-glial antigen 2 (NG2) proteoglycan [16,17,18]. PDGF-A is the most potent OPC mitogen and survival factor, which is produced by both astrocytes and neurons; consequently, overexpression of this growth factor, e.g., during development, leads to increased OPC number [19]. When pre-OLs engage with a target axon, they lose bipolarity to acquire a ramified morphology and start to build filamentous myelin outgrowths [8]. At this differentiation stage, pre-OLs are characterised by the expression of three main myelin-associated markers, 2′, 3′-cyclic-nucleotide 3′-phosphodiesterase (CNPase) and the cell surface markers O4 and O1 [20]. O4 is already expressed in late progenitors, whereas O1 is typical of pre-myelinating OLs [21]. Mature, differentiated OLs are characterised by the production of myelin and myelin-related proteins, such as myelin basic protein (MBP) [22], which is expressed on the cytoplasmic surface of the plasma membrane [23], the transmembrane protein myelin proteolipid protein (PLP) [24] and myelin associated glycoprotein (MAG) [25].

## 3. Voltage-Gated Channels in Oligodendroglial Cells and Myelination

OPCs receive synaptic inputs from neurons [26] and express voltage-gated ion channels (such as Na^+^ and K^+^ channels; Figure 2) and various neurotransmitter receptors [27,28,29,30,31]. During motor learning in mice, a rapid differentiation of OPCs in the motor cortex was found, followed by a subsequent increase in compensatory proliferation to return to homeostatic OPC density [32]. Moreover, it was recently demonstrated that motor learning could enhance oligodendrogliogenesis and remyelination after cuprizone (CPZ) intoxication, a mice model of demyelination [33]. Similarly, oligodendrogliogenesis and *de novo* myelination were increased following spatial learning in the water maze test [34]. Social experienced after cuprizone treatment also influenced remyelination in adult mice [35] and oligodendrogliogenesis in adolescent mice [36]. These studies imply that activation of neuronal circuits plays an important role in white matter development [37]. By speeding AP conduction, myelination increases the brain’s cognitive abilities. During development or learning, a re-arrangement of myelin thickness or internode length may help to tune the speed of conduction along myelinated axons [38]. This can promote synchronous neuronal firing [39], make impulse propagation time less dependent on the spatial trajectory of the axon transmitting information between areas [40], or adjust propagation delays along the cochlear nerve to promote sound localization [41]. Magnetic resonance imaging (MRI) reveals changes in white matter microstructure, perhaps reflecting alterations of myelin, when human subjects learn a skilled motor task such as playing piano [42] or juggling [43]. Together, these studies suggest that adaptive myelination is a physiological and essential aspect of neural plasticity and reveal the presence of an important crosstalk between neurons and OLs.

Among demyelinating pathologies, multiple sclerosis (MS) represents the most common chronic disease in the CNS. MS is primarily considered an immune-mediated disease and is characterized by focal areas of infiltrated lymphocytes causing neuroinflammation with associated demyelination that spreads over the brain and spinal cord with time [44]. The classic pathological hallmark of MS was long considered to be the presence of focal white matter demyelinating lesions. However, pathological changes are also detectable in normal-appearing white matter (NAWM), as well as in the CNS grey matter, with the presence of focal grey matter lesions and grey matter atrophy [45]. Indeed, it can be supposed that one of the mechanisms involved in the pathogenesis of MS is the altered gene expression and reorganization of ion channels at the level of demyelinated axons [46].

It is now known that ion channels are considered an important target class for the study and treatment of various pathologies. In particular, the Food and Drug Administration (FDA) has recognized both ligand-dependent and voltage-dependent ion channels among the top pharmaceutical targets of approved agents [47]. Therefore, ion channel dysregulation on neurons and glial cells can probably contribute to axonal degeneration during chronic inflammation in the brain and spinal cord [46,48] and to abnormal activation in immune cells. These hypotheses are supported by numerous studies where ion channel blockers have been shown to modulate CNS damage and symptoms due to experimental autoimmune encephalomyelitis (EAE), a mouse model for MS. Indeed, different studies on EAE mice revealed distinct ion channel families as key players in pathophysiological processes of demyelination and could be important drug targets for this disease [49,50].

According to their vast expression, ion channels have the potential to influence nearly every stage of MS pathogenesis. The regulation of the immune response is, among other modulators, dependent on ion channels which allow peripheral T lymphocytes to proliferate and to produce inflammatory cytokines [51]. 

Within the class of drugs targeting ionic channels on immune cells is Glatiramer Acetate (GA), an immunomodulatory drug used in the treatment of MS, which seems to play an important role on B lymphocytes, where it modifies both the immune response and the activation of ion channels (see Table 1). In the first case, GA inhibits B lymphocytes maturation, with a concomitant increase in the number B cell precursors and/or naïve B cells. In the second case, it was observed that GA was able to modulate Ca^2+^ homeostasis in these cells. Indeed, it was observed that GA changed the expression of K^+^ and Cl^−^ channels but, also, Ca^2+^ release activated Ca^2+^ entry and transient receptor potential (TRP) channel opening [52].

A bulk of evidence indicates that the application of ion channel blockers, such as those for Na^+^, Ca^2+^ and K^+^, greatly improves EAE disease course and delays the pathology onset after immunization in comparison to sham-treated control mice. Based on above evidence, ion channels, expressed either on central or peripheral cells, are regarded as putative promising new targets for MS, particularly in the progressive form of the disease. Indeed, the beneficial effect of ion channel blockers observed in pre-clinical studies may be mediated by two distinct pathways: modulation of an ion channel on nerve cells, which may facilitate neuronal survival, as well as inhibition of ion channels on T cells, which may provide immunomodulation [78].

Currently, clinical translation of ion channel-targeting compounds remains a major challenge in MS. Indeed, only a few clinical trials have been performed and most approaches are still at early preclinical stage, possibly due to the lack of selective compounds, on one side, and poor knowledge of their exact role in the context of autoimmunity, on the other [78].

### 3.1. Voltage-Gated Na^+^ Channels in Oligodendroglial Cells

As summarized in Table 2, TTX-sensitive voltage-gated Na^+^ channels (Nav) were found on A2B5^+^/NG2^+^ OPC [28,79,80,81,82,83,84], but their expression is restricted to the earliest stages of OPC differentiation, being down-regulated during maturation into OLs [85,86,87]. Nav have also been found in the white matter of cortical slices obtained from P3-P8 animals, but not at later ages (P10-P18) [83,86]. In confirm, Paez and colleagues investigated Na^+^ currents in oligodendroglial cells recorded from brain slices of transgenic mice expressing green fluorescent protein (GFP) under the control of the PLP gene promoter (OL-GFP) [86,88]. In these transgenic OL-GFP animals (P4-P6), the majority of O4^+^ cells (91%) in the *corpus callosum*, representing immature pre-myelinating OLs, have spherical cell bodies with multiple short radiating processes and lack inward Na^+^ currents [86,89]. By contrast, most of Sox9^+^ cells (90%) close to the lateral ventricles have simple morphology with few processes and expressed abundant Nav [86]. 

Importantly, some Nav-expressing OPCs can generate APs when stimulated by depolarizing current injection [85,94,124,125]. However, OPC-evoked AP has a higher threshold, smaller amplitude and slower kinetics than neuronal counterpart; since synaptic inputs produce only minimal depolarization in NG2^+^ cells, the physiological relevance of OPC excitability is unclear [110]. However, APs may be involved in a mechanism through which OPCs sense electrically active axons in the environment [94]. In confirm, NG2^+^ cells receive glutamatergic input upon stimulation from CA3 Schaffer collateral axons in the CA1 region of the hippocampus [28] and, similarly, in the *corpus callosum* from axons coursing through this region [126,127]. Synaptic input- and/or Na^+^ channel-mediated electrical activity may serve as a signal between unmyelinated axonal sections and OPCs that are ready to differentiate into OLs to myelinate these axonal targets [94]. However, not all NG2^+^ cells with Na^+^ currents appear to fire regenerative APs, possibly because of low Nav densities [28,95,124]. Hence, Na^+^ channels are likely to be active upon depolarization in NG2^+^, probably contributing to the high level of proliferation [95], and could play a role in development and/or CNS repair, e.g., migration. Table 2 summarizes Nav expression and functions in oligodendrogliogenesis.

#### Nav Channels in Demyelinating Diseases

In healthy conditions, Nav1.2 is expressed on immature, unmyelinated neurons, where it supports impulse transmission. However, this channel subtype is no longer detectable in adult mice myelinated neurons, where the closely related Nav1.6 channel supports saltatory conduction of nerve impulses at nodes of Ranvier [128]. Nav1.2 channels produce rapidly activating and inactivating currents and appear to support action-potential conduction, which occurs before myelination. Nav1.6 channels, on the other hand, produce a persistent current that follows the rapidly activating and inactivating currents, and Nav1.6 is co-expressed together with the Na^+^/Ca^2+^ exchanger (NCX) along degenerating axons in EAE [93,129]. This event leads to sustained depolarization causing an intracellular Ca^2+^ overload that triggers mitochondrial dysfunction and leads to neuronal cell death [90,129]. During demyelination, Nav1.2 and Nav1.6 are found to be overexpressed at demyelinated sites of the axonal membrane in EAE models [90]. Thus, redistribution of Na^+^ channels might represent an initial compensatory process preserving conductance and axonal integrity but, at later stages, accelerates chronic neurodegeneration [46]. Accordingly, in post-mortem tissues from patients with secondary progressive MS, both Nav1.6 and Na^+^-Ca^2+^ exchanger (NCX) colocalize with amyloid precursor protein, a marker of axonal injury [93]. Of note, data demonstrated that NCX3 is strongly upregulated during oligodendrocyte development [130] and its activity is required for the maintenance of Ca^2+^ transients necessary to myelin synthesis in oligodendrocyte cultures [131]. Recent evidence confirmed this hypothesis and deepened knowledge in the field by demonstrating that neuronal activity, presumably in form of local extracellular [K^+^] changes, might locally influence NCX-mediated Ca^2+^ transients in OLs thus triggering local MBP synthesis in the vicinity of an active axon [132].

Furthermore, an abnormal upregulation of mRNA and protein for Nav1.8 was found within cerebellar Purkinje cells in EAE mice and MS humans [91]. This suggests that abnormal electrical signalling within the cerebellum due to a transcriptional channelopathy could contribute to MS [91]. In confirm, the Nav1.8 subtype in EAE mice is found to be aberrantly expressed in Purkinje neurons and Nav1.8-selective blockers can partially reverse EAE-induced neurological deficits, including abnormal Purkinje cell firing [53]. Indeed, the new orally bioavailable Nav1.8-selective blocker PF-01247324 improved motor coordination and cerebellar-like symptoms in transgenic mice expressing Nav1.8 in Purkinje neurons and in wild-type mice with EAE [53].

Wang et al. found that the expression of Nav1.5 was significantly increased in reactive astrocytes within MS lesions, while Nav1.1-Nav1.3 and Nav1.6 exhibited only minor changes [92], thus suggesting a role of Nav in the functional response of reactive glial cells in MS lesions. In accordance, it was found that the Nav blocker phenytoin can attenuate microglial activation, migration and phagocytosis [56,133], as well as the release of IL-1 and TNF-α proinflammatory mediators, in EAE lesions [133]. Several studies have also shown that Nav blockers can reduce axonal injury and improve the clinical score of MS patients [56,134]. Indeed, safinamide, at both high and low doses, significantly protected against neurological deficit as well as microglial activation and increased the pool of protective M2 type microglial cells in EAE mice [54]. In cultured microglia, safinamide suppressed superoxide production via inhibition of NADPH oxidase, an enzyme which is primarily responsible for the generation of reactive oxygen species. As these effects of safinamide were comparable with that of the cognate Na^+^ channel inhibitor flecainide, the mechanism behind safinamide mediated protection was attributed to the inhibition of Navs [54,55].

In confirm, the use of Nav blockers, such as flecainide, phenytoin and, more recently, lamotrigine preserved both axonal integrity and electrical conduction in EAE mice. Importantly, these agents maintained conduction along the CNS and reduced disability scores in treated animals [56,57]. Phenytoin was also tested in a phase II clinical trial on 86 participants affected by acute demyelinating optic neuritis, a common symptom of MS, and has proved to be neuroprotective [58,59]. On the other hand, lamotrigine, also tested in a phase II clinical trial on 120 patients with secondary progressive MS, did not affect the rate of changes in partial cerebral volume over 24 months vs. placebo. However, a relevant side effect was pointed out by the study as a reduction in white matter volume was detected in patients during the first 12 months after lamotrigine administration, an effect that was however lost upon discontinuation of treatment [58,60]. Finally, paroxysmal dysarthria and ataxia in patients with MS can respond to treatment with Na^+^ channel blockers such as carbamazepine [61].

Although encouraging preclinical data, Nav-targeting compounds still miss clinical translation due to rebound effects after discontinuation of the drug. Only phenytoin differs from related compounds as both preclinical and clinical studies unveiled its neuroprotective properties, based on Nav blockade [135]. For a summary on the effects of Nav-targeting compounds in demyelination see Table 1.

### 3.2. Voltage-Gated Ca^2+^ Channels in Oligodendroglial Cells

Ca^2+^ signalling has been shown to regulate many oligodendroglial cell functions, including proliferation, migration, process extension, differentiation, and myelination [86,96,110,136,137,138,139]. Intracellular Ca^2+^ can increase in NG2^+^ cells throughout several mechanisms, such as direct influx through plasma membrane voltage-gated Ca^2+^ channels (Cav) or ligand-gated channels [140,141], as mentioned below, or by the release from internal Ca^2+^ stores [142,143] as well as Ca^2+^-permeable acid-sensing ion channel opening [144].

Cav immunoreactivity was found in CNS white matter, located in oligodendroglial soma, projections and paranodal wraps [145] (see also Table 2). Electrophysiological recordings demonstrated the functional expression of low-voltage and high-voltage activated Ca^2+^ currents in OPCs from *corpus callosum* and mouse cortex [81,83,96,97], with pharmacological and voltage-dependent properties typical of T-type and L-type Cav, respectively [97]. Indeed, several studies have reported that Ca^2+^ currents through Cav appear to diminish with maturation of OLs from progenitors to mature cells in culture [83,104].

These electrophysiological results have been confirmed by RNA-sequencing transcriptome database that revealed high expression of L-type Ca^2+^ channel subunits in OPCs and subsequent downregulation in newly formed and myelinating OLs [98], indicating that Cav plays a role during the first steps of OPC maturation. The presence of Cav was also confirmed by mRNA analysis for L-type (Cav1.2 and Cav1.3), T-type (Cav3.1 and Cav3.2), P/Q-type (Cav2.1), and N-type (Cav2.2) α1 subunits in NG2^+^ cells [110]. In particular, Cav1.2 represents the primary pore-forming subunit in OPCs, as siRNA knockdown of Cav1.2 in NG2^+^ cells removes ∼75% of the Ca^2+^ elevation following depolarization [96].

Functional Cav are reported to be necessary for glial cell migration in vivo during olfactory glomeruli formation in the developing antennal lobe of sphinx moth *Manduca sexta* [105]. In addition, in deafferented antennal lobes in which glial cells fail to migrate [106], glial Cav currents are absent, indicating that Cav in glial cells are required to induce or maintain the migration of antennal lobe glial cells into the developing neuropil of the moth [105].

In addition, data indicate that L-type Cav activation, probably mediated by PDGF, contributes to spontaneous Ca^2+^ oscillations in the OPC soma, leading to accelerated migration and process formation [104]. Furthermore, an increase in amplitude and frequency of Ca^2+^ transients is one of the mechanisms underlying AMPA-induced stimulation of OPC migration [107], as described below. Furthermore, Ca^2+^ transients may affect the recycling of cell-adhesion receptors and induce the rearrangement of cytoskeletal components, which are essential for cell movement [108].

When OPCs are grown in high extracellular K^+^, used as a depolarizing stimulus to activate Cav, they are prompted towards maturation, as demonstrated by a more complex morphology and a significant increase in the expression of mature markers [109]. At the same time, blocking the expression of the Cav α1.2 subunit, that conducts L-type Ca^2+^ currents, significantly prevents OPC culture maturation [96]. Accordingly, Cav1.2 deficient OPCs present inhibited proliferation and disruption of proliferative response to PDGF, the best known and most active mitogen for OPCs [96]. Interestingly, it was shown that store-operated Ca^2+^ entry, as well as Ca^2+^ release from intracellular stores, are essential mechanisms for PDGF-mediated mitotic action in OPCs [86]. A widespread hypothesis is that Ca^2+^ entry by L-type channels modulates OPC division and cell maturation through independent intracellular pathways. Cav seem to be essential for cell cycle progression of in mitotic OPCs whereas, in post-mitotic pre-OLs, the same channels are playing an important role in cell maturation. In support to this hypothesis, it was demonstrated that a loss of Cav1.2 in oligodendroglial cells affects axonal contact in co-cultured cortical neurons and consequently inhibits the initial steps of myelination [96]. Moreover, deletion of Cav1.2 in OPCs reduces OL maturation and myelination in the postnatal mouse brain and impairs remyelination in a CPZ model [100,101].

Importantly, it is likely that factors involved in physiological myelination also participate in remyelination of the injured CNS. In this regard, a significant increase in the activity of OPC L-type Cav was found in demyelinated *corpus callosum* of CPZ-treated mice, suggesting that these channels may play a key role in the induction and/or survival of newly generated OPCs after an insult [102]. Cav expression and functions in oligodendroglial cells are summarized in Table 2.

#### Cav Channels in Demyelinating Diseases

Several evidences support the notion that aberrant Cav-mediated currents contribute to the pathophysiology of MS or EAE. Increased Ca^2+^ influx through Cav was assumed to facilitate neurological impairment and histological damage in EAE mice and, by inference, MS [146]. Pregabalin (Lyrica^®^) is prescribed to MS patients to treat neuropathic pain by targeting Cav [62] and, in addition, it could provide neuroprotection by inhibiting exaggerated Cav currents during excitotoxicity and neuroinflammation. Indeed, it has been demonstrated that Pregabalin treatment alleviates EAE symptoms in mice possibly by reverting, at neuronal level, intracellular Ca^2+^ overload in EAE lesions [63]. However, in the same paper, the authors pointed to a significant reduction of hippocampal long-term potentiation in pregabalin-treated EAE mice, thus warning of potential side effects on memory and learning processes [58,63]. In accordance with deleterious effects on memory, two recent clinical studies showed that perioperative pregabalin reduced spatial working memory in humans [64] and its misuse led to cognitive impairment [65]. 

An abnormal redistribution of N-type Ca^2+^ channels was found in acutely injured axons, followed by rearrangement of the axonal membrane after injury [147]. Since Nav are known to redistribute along demyelinated axons [46], a similar mechanism may also exist for Cav. Additionally, the N-type Cav2.2 was detected also on mature OLs [99] and the expression of the pore forming α1B-subunit of Cav2.2 was found in MS and EAE plaques and was overexpressed in active lesions [103].

In mice, it has been shown that, after knockout of Cav1.2, axonal myelination is inhibited and OPC maturation disturbed [100]. However, the L-type Ca^2+^ channel (Cav1.2, Cav1.3, Cav1.4) blockers Bepridil and nitrendipine had comparable beneficial effects in reducing neuroinflammation and axonal pathology on EAE mice [66]. When nimodipine was administered preventively at the time point of disease induction, EAE severity and demyelination decreased [67]. Recently, nimodipine has been reported to have positive effects on Schwann cells, astrocytes and neurons, being associated to increased phosphorylation of either protein kinase B and the cyclic adenosine monophosphate response element-binding protein (CREB) [68,148]. It is known that axonal Ca^2+^ overload activates the Ca^2+^-dependent protease calpain, leading to disruption of the cytoskeleton and to other structural and functional alterations of the axon [149]. Of note, nimodipine also downregulated the expression of calpain as well as the pro-apoptotic protein caspase 3, whereas calbindin expression was upregulated, indicating that modulation of Ca^2+^ homeostasis and prevention of intracellular Ca^2+^ overload might be responsible for the neuroprotective properties of this Cav blocker [68,69]. 

Interestingly, recent studies have reported that the Cav1.2 channel blockers nimodipine and verapamil exert their neuroprotective effects through anti-inflammatory properties [70], possibly preventing microglial activation [71] and down-regulating TNFα and IL1β expression in the hippocampus [72,73]. However, microglial cells do not express functional Cav1.2 channels [150], thus the anti-inflammatory effects of these drugs are likely mediated by their block on other cell types [72]. Recently, it was found that animals injected with nimodipine during CPZ-induced demyelination displayed a reduced astrocyte and microglia activation and proliferation as well as a faster and more efficient brain remyelination [74]. Cav1.2 channels are not present in mature OLs [96,104], but they are expressed by OPCs where they are essential for maturation [100,101], as mentioned above. This suggests that reducing Cav currents inhibits astrocyte and microglia activation during demyelination. Consequently, pool of proliferating OPC increases as well as the number of myelinating OLs, leading to a beneficial effect for myelin regeneration [74]. However, deletion of the Cav1.2 channels in GFAP^+^ astrocytes did not prevent myelin damage during CPZ treatment. 

Hence, it appears that Cav blockers might represent promising targets for demyelinating diseases as they concur to pathological intracellular Ca^2+^ overload in neurons and immune cells. Nevertheless, there are no trials for clinical translation of Ca^2+^ channel blockers so far [151]. For a summary on the effects of Cav-targeting compounds in demyelination see Table 1.

### 3.3. Voltage-Gated K^+^ Channels in Oligodendroglial Cells

The OPC resting membrane potential (V_rest_) is near the calculated equilibrium potential for K^+^ (EK), i.e., −80 mV, suggesting that K^+^ channels account for the majority of the resting membrane conductance. The predominant K^+^ channel subtypes open at rest (‘leak’ channels) are the inward-rectifier Kir4.1 and two-pore (K2P) K^+^ channels [110]. The RNA-Seq transcriptome database shows that Kir4.1 mRNA is expressed at high levels in NG2^+^ cells [98] (see Table 2). Kir4.1 mediates inward currents observed in NG2^+^ whole cell recordings upon membrane potential hyperpolarization lower than –100 mV. These currents are blocked by low (200 µM) concentrations of extracellular Ba^2+^, an inhibitor of Kir channels [110,114,115]. Kir4.1 facilitates clearance of extracellular K^+^ released during axonal firing, thus maintaining resting membrane potential and AP propagation [46]. Deleting Kir4.1 in mice causes impaired OL maturation and myelination during development, leading to neuronal degeneration [115] and selective deletion of Kir4.1 from OPCs or mature OLs also results in profound functional impairment and axonal degeneration [114,115].

Outward rectifying voltage-gated K^+^ channels (Kvs) are also prominent in OPCs, where they are known to regulate cell proliferation and differentiation, and are subsequently downregulated during differentiation [87,123] (see Table 2). Upon depolarization, NG2^+^ cells display a non-linear current-to-voltage profile that is shaped by the activation of A-type (I_A_) and delayed-rectifier (I_K_) K^+^ channels. These currents have been extensively characterized in OPCs recorded from cell cultures or in brain slice preparations [79,80,81,82,117,118]. When challenged by a depolarizing voltage step, 4-aminopyridine (4-AP)-sensitive I_A_ contribute to the initial ‘peak’ outward current during the depolarizing phase, due to their rapid activation and inactivation kinetics. Moreover, TEA-sensitive I_K_, which activate more slowly and do not inactivate, contributes to the sustained ‘steady-state’ current of the depolarizing stimulus [79,80,81,82,117,118]. The relative proportion of the two current components varies by the region of origin. When compared to cortical NG2^+^ cells, white matter OPCs in P5-P10 mice present higher I_K_ current densities, while I_A_ density is comparable, resulting in a higher I_K_/I_A_ ratio [124]. During maturation, outward K^+^ conductances, in particular I_K_, in OPCs undergo a strong downregulation up to almost completely disappearance in mature OLs [80,87]. In parallel to I_K_ downregulation, there is a gradual increase in the expression of Kir, that represents the main conductance observed in mature OLs [111], as demonstrated by the Ba^2+^-sensitivity of overall OPC conductance increases during maturation [152]. 

RT-PCR and immunocytochemical localization in cultured NG2^+^ cells have shown robust expression of Kv1.2, Kv1.3, Kv1.4, Kv1.5, Kv1.6 and Kv7.2 mRNA and protein, with Kv1.5 and Kv1.6 showing highest levels among Shaker-type delayed rectifiers [119,120]. In addition, mRNA for several non-Shaker delayed rectifier channels, Kv7.2 and Kv2.1, are also abundantly expressed [110]. Kv1.4, the only Shaker-type channel to display A-type properties, presents low expression in NG2^+^ cells database [110]. Other A-type channel subunits that are greatly expressed in OPCs are Kv4.2, Kv4.3, and Kv3.3. Kv1.3 is upregulated during the G1 phase of cell cycle, and blockade of this channel with specific toxins prevents G1/S transition [120]. Conversely, overexpression of Kv1.3 or Kv1.4 promotes OPC proliferation in the absence of mitogens, while overexpression of Kv1.6 inhibits proliferation in the presence of mitogens [123]. On the other hand, neither knockdown nor overexpression of Kv1.5 affect OPC proliferation [119,123]. Of note, and differently from cell proliferation, differentiation of cultured NG2^+^ cells into OLs is not significantly affected by overexpression of Kv subunits but is impaired by the Kv blocker TEA [87,109], demonstrating that oligodendroglial cell proliferation and differentiation might be differently regulated [123].

The large conductance Ca^2+^-activated (BK) channel KCa1.1 is highly expressed in NG2^+^ cells [110]. This confirms previous findings that BK channels, which are both voltage- and Ca^2+^-dependent, are expressed in cultured NG2^+^ cells [153]. Other K^+^ channels may also be important in maintaining oligodendroglial cell functions and integrity, including Kir2.1, Kir7.1 and TASK1 channels [111,112,113]. For a summary on Kv and Kir expression in oligodendroglial cells, see Table 2.

#### Voltage-Gated K^+^ Channels in Demyelinating Diseases

As mentioned above, deleting Kir4.1 channels in mice causes impaired myelination and OL maturation during development, leading to neuronal degeneration [115]. In confirm, serum levels of antibodies against Kir4.1 were enriched in MS patients, suggesting that Kir4.1 could be a target of antibody responses in this pathology [116]. 

In myelinated axons, Kv1.1 and Kv1.2 are located underneath the myelin sheath in the paranodes or internodal regions [46], but, after demyelinating insults, alterations in K^+^ channel expression and distribution along the axon are reported. Indeed, a reduction of Kv1.2, Kv1.4, and Kv2.1 has been demonstrated in EAE mice, correlating with disease severity [122]. Moreover, Calvo et al. found that, although Kv1.1 and Kv1.2 expression levels decrease, they redistributed form the juxtaparanode into the paranode in a spinal nerve transection (SNT) model of neuropathic pain [154]. In contrast, Kv1.4 and 1.6 expression increases within justaparanodes and paranodes [154]. These findings are consistent with previous studies reporting a Kv mislocalization in EAE animals and in post-mortem human MS lesions [121]. In accordance, administration of a Kv1.1 selective blocker (BgK-F6A) ameliorates disease course in EAE mice [75], and treatment with the I_A_ blocker 4-AP enhances axonal conduction [76]. As I_A_ is widely diffused along demyelinated axons and contributes to MS symptoms [155], in 2010 dalfampridine, an extended-release form of 4-AP, has been approved by the FDA to improve walking in MS patients [77]. The mechanism by which Kv blockers could exert neuroprotection is ascribed to the block of excessive K^+^ outflow from axons causing extracellular K^+^ homeostasis outbalance that postpones the K^+^ reversal potential to more positive values. Consequently, a gradual depolarization of neuronal membrane occurs with a consequent inactivation of Nav leading to impaired action potential propagation. Hence, Kv block may help preserving signal conduction. Furthermore, as axonal K^+^ outflow leads to intracellular K^+^ depletion causing water loss and disinhibition of proapoptotic enzymes [156], these compounds might also provide neuroprotection by preserving intracellular K^+^ homeostasis. Of note, a recent paper demonstrated that, although Kv1.4 deficiency decreases OPC proliferation in vitro, it does not influence de- or remyelination in the CPZ model [157]. Moreover, in the same study, Kv1.4 deficiency leads to an ameliorated course of EAE and results in reduced Th1 responses. These data argue for a peripheral effect of Kv1.4 on immune cells, possibly *via* glial cells. Since the authors did not observe any change in the CPZ model, it is tempting to speculate that the benefits observed in the EAE model are not only due to remyelination, but also to a reduced impact of immune system on the CNS, as discussed previously [158,159]. For a summary on the effects of Kv-targeting compounds in demyelination see Table 1.

## 4. Neurotransmitters in Oligodendroglial Cells and Myelination

OPC proliferation and myelination are coordinated by communication with axons and neurons within the circuit [160,161,162,163]. It has recently been suggested that OPCs are able to communicate with surrounding neurons through non-synaptic junctions during OPC maturation to myelinating OLs [164]. It is known that growth factors are involved in this axon-OL crosstalk; in addition, OPCs receive excitatory and inhibitory synaptic inputs mediated, respectively, by glutamate and GABA [27,28,165], thus suggesting that these neurotransmitters may also control OL development and myelination [166,167,168]. Co-transmitters and neuromodulators like ATP and adenosine are also important mediators for oligodendrogliogenesis [169,170]. Although OLs are able to produce myelin in the absence of neuronal activity [171], an in vitro study demonstrated that OLs preferentially myelinate electrically active axons [172]. However, to date the molecular mechanisms underlying OPC proliferation and myelination by neurotransmitters/neuromodulators are only partially known (see Table 3 for a summary).

### 4.1. Glutamate

Increasing evidence shows that glutamate modulates OPC proliferation, migration and myelination during development [27,193]. Since the activation of glutamate receptors leads to intracellular Ca^2+^ increase, Ca^2+^ signalling has also been postulated to participate to OPC development, as described above [194,195,196]. 

Initial in vitro studies suggested that glutamate α-amino-3-hydroxy-5-methyl-4-isoxazolepropionic acid and kainate receptors (AMPAR and KAR) block prevents OPC proliferation and maturation by inhibiting K^+^ channel activity [109,173,174]. Accordingly, AMPAR activation in an in vivo remyelination model promotes myelination [176,177], whereas its inhibition in OPCs leads to enhanced cell proliferation [197]. Indeed, the role of glutamate on OPC proliferation appears controversial as Fannon et al. demonstrated that AMPAR antagonism, but not N-Methyl-d-aspartic acid receptors (NMDAR) antagonism, increases OPC proliferation in organotypic cerebellar slices [175], a result that has not been reproducible by Hamilton et al. [179]. Intriguingly, a recent paper demonstrated that axons regulate oligodendrogliogenesis via the AMPAR subunit GluA2, since modifying this subunit decreases OPCs differentiation, but not proliferation [198].

Glutamate released by excitatory neurons may also serve as a chemoattractant, stimulating OPCs migration toward their target destination in the developing brain [107]. Indeed, activation of glutamate receptors on OPCs accelerates integrin mediated OPC motility via mechanisms that involve AMPARs [107]. 

Furthermore, it has been demonstrated that the molecular mechanisms underlying glutamate signalling in oligodendrogliogenesis involve elevated levels of the cell cycle inhibitors p27^Kip1^ and p21^Cip1^ [4], whose expression is usually high during cell cycle. Thus, p27^Kip1^ and p21^Cip1^ represent an “internal clock” or timing component of OL lineage cell progression, leading to stalling at the G1-S phase transition by dissociating cyclin-cdk complexes [199]. 

NMDARs, probably located extrasynaptically, are enriched on cell processes and myelin sheaths of OLs [200], which makes their detection difficult by whole-cell patch clamp recordings. This NMDAR distribution strengthens the importance of a neuro-glial communication between OL lineage cells and axons [201]. However, the role of NMDARs in OPC development and myelination is controversial [194,202]. Although in vitro studies have demonstrated a positive effect of NMDARs on OPC migration and differentiation [178], De Biase et al. found that NMDARs are not required for oligodendrogliogenesis and myelination, but they may regulate the AMPAR signalling [203]. Moreover, it has been hypothesized that NMDARs provide metabolic support to myelinated axons [10,200]. Given these controversial data, further studies are needed to clarify whether NMDARs may play a role in myelination in vivo.

The complexity of molecular mechanisms involved in myelination reported here highlights how glutamate signalling in OPCs may depend on developmental stage and brain area.

### 4.2. GABA

Oligodendroglia express GABA_A_ and GABA_B_ receptors (GABA_A_Rs and GABA_B_Rs), while no data are present to date about a role of the third GABAR subtype, the ionotropic GABA_C_R [204].

Zonouzi et al. observed that a decreased GABA_A_R signalling to NG2 cells in the hypoxic brain results in increased OPC proliferation, delayed OL differentiation and causes dysmyelination in mice [165]. Similar effects were reported by the intraperitoneal injection of the GABA_A_R antagonist bicuculline whereas, conversely, blockade of GABA catabolism or uptake reduced NG2 cell numbers and increased the formation of mature OLs both in control and hypoxic mice [165]. These results pointed to a pro-myelinating effect of GABA_A_R activation that has been recently confirmed by Cisneros-Mejorado using magnetic resonance imaging in a rat demyelination model consisting in ethidium bromide (EB) injection into the cerebellar peduncle [180]. On the other hand, Hamilton and colleagues demonstrated that endogenously released GABA, still acting on GABA_A_Rs, reduced the number of OL lineage cells in organotypic cerebellar slices but also inhibited myelination in this limited in vitro model [179]. This discrepancy in the role of GABA_A_R in myelination could be attributed to in vivo [165,180] vs. in vitro [179] experimental designs. 

In contrast to GABA_A_R, whose sustained expression depends on the interaction with axons and thus in isolated OL cultures is downregulated during the differentiation process [83,205], OLs maintain the expression of metabotropic GABA_B_Rs over time [182,206]. Interestingly, previous findings demonstrated that treatment with the GABA_B_R agonist baclofen leads to an increase in proliferation and migration by decreasing cAMP levels in cultured OPCs [181]. Moreover, exogenous GABA increases myelination and MBP expression in DRG-OPC co-cultures, suggesting that GABA promotes myelin formation when OLs are in contact with axons [182,206]. In the same study, in rat OPC cultures, GABA and the specific GABA_B_R agonist baclofen stimulates OPC differentiation by increasing OL process branching and myelin protein expression via Src phosphorylation [182]. 

Overall, it appears that GABA_A_Rs and GABA_B_Rs exert opposite roles on OPC proliferation, whereas both receptors enhance myelination. 

### 4.3. Purines

Purinergic signalling plays an important role in OL lineage cells both in development and in re-myelination during homeostatic and allostatic conditions [113,169,207]. Purine receptors are basically classified in P1 receptors (P1R), that bind adenosine, and P2 receptors (P2R), that bind ATP, ADP and purine nucleotides-conjugated sugars, such as UDP-glucose [208]. Up to date, two subfamilies of P2Rs were identified, P2X receptors (P2XRs), ligand-gated ion channels, and G protein-coupled P2Y receptors (P2YRs). Regarding P1 adenosine receptors, four different G protein-coupled receptors (A_1_Rs, A_2A_Rs, A_2B_Rs and A_3_Rs) were identified [209].

P2XRs, further divided into P2X_1-7_, are expressed by OPCs and OLs, with most robust evidence for the P2X_7_R subtype [210]. P2X_7_R activation mediates a rise in intracellular Ca^2+^, through direct influx by the channel pore, and activates multiple intracellular pathways, including MAPK, PKC, and PI3K, all involved in the regulation of OPC proliferation, differentiation and myelination. In addition, P2X_7_R also participates to OL damage and myelin loss during ischemia or neuroinflammation [184,185]. The facilitating role of P2X_7_R in OPC migration has been outlined by Feng et al. who demonstrated that high concentrations of ATP or the P2X_7_R agonist BzATP increased the number of migrating OPCs in vitro, an effect abolished by pre-treatment with the P2X_7_R antagonist oxidized ATP [183]. However, as known for this P2R subtype [211], exaggerated P2X_7_R stimulation by extracellular ATP causes intracellular Ca^2+^ overload resulting in EAE-induced OL death [186].

P2YRs, further divided into P2Y_1,2,4,6,11,12,14_ [208], also play an important role in OPCs and OLs. In particular, the prominent expression of P2Y_1_R subtype mediates intracellular Ca^2+^ raise, through G_q_-mediated inositol-phosphates pathway, which facilitates OPC migration and affects proliferation and differentiation [187]. Indeed, it has been demonstrated that ATP and ADP, by acting through P2Y_1_Rs, induced OPC migration and inhibited OPC proliferation in purified cell cultures and in cerebellar tissue slices, and all these effects were blocked by the P2Y_1_R antagonist MRS2179 [188]. In addition, P2Y_12_R, known to be highly expressed in OLs, is strikingly downregulated in the cerebral cortex of *post-mortem* MS brains and, importantly, decreased P2Y_12_R immunoreactivity in proximity to the lesions directly correlates with the extent of demyelination [113,189]. Of note, the recently deorphanized P2Y-like receptor GPR-17, activated by the uridine nucleotide-conjugated sugar UDP-glucose, is emerging as a key regulator of oligodendrogliogenesis as it enhances OPC migration [87] and differentiation [87,191] in preliminary phases of OPC-preOL maturation. On the other hand, at later stages, i.e., CNPase^+^/O1^+^ cells, its overexpression impairs final OL differentiation and myelination in vitro [212]. Of note, recent data demonstrated that GPR17 is overexpressed in active lesions and in NAWM of *post-mortem* MS brains [190]. 

Regarding adenosine, all four subtypes of G protein-coupled adenosine receptors have been identified in OPCs, where they regulate migration, proliferation, and differentiation [170]. One mechanism by which adenosine affects OPCs development is voltage-dependent K^+^ channel regulation, as these channels are known to control cell cycle progression and/or migration by regulating membrane potential and cell volume [87,109,213], as stayed above. In particular, a dual role of adenosine has been reported in modulating OPC development. By stimulating A_1_Rs, adenosine inhibits OPC proliferation and enhances their differentiation and axonal myelination in OPC-DRG co-cultures [192]. However, by acting on A_2A_Rs [117] and/or A_2B_Rs [118], it inhibits OPC differentiation by decreasing TEA-sensitive Kv currents necessary for the expression of mature OL markers [109,117,118,170,214], while no effects on cell proliferation were found [118]. The A_3_R subtype has been reported to induce OL apoptosis [170].

Overall, the effects mediated by adenosine on oligodendrogliogenesis and myelination may depend on the developmental stage or by the environmental conditions during demyelinating pathologies. However, due to the important actions of adenosine on I_K_ currents and thus on OPC maturation, brain A_1_R, A_2A_R and A_2B_R might represent new pharmacological targets for demyelinating pathologies such as MS, stroke and brain trauma.

## 5. Conclusions

Taken together, data indicate ion channels--expressed either on neurons or glia--play a key role in demyelinating pathologies such as MS. Regarding oligodendrogliogenesis, ionic channel expression changes during maturation [27,28,29,30,31]. OPC express Nav, Cav and Kv, that are decreased during maturation [80,83,85,86,87,104,117], while N-type calcium channels (Cav2.2) [99] and Kir (Kir4.1) appear [152]. 

During demyelinating insults, this tuning of ion channel expression is dysregulated. In MS and EAE lesions, an overexpression of Nav1.2, 1.6, 1.8 [53,90,91] was found and, consistently, different Nav blockers proved effective in preventing neurological and histological damage in preclinical models. However, when translating to the clinic, only phenytoin proved effective in phase II clinical trials for demyelinating optic neuritis [58,59], while carbamazepine partially ameliorated paroxysmal dysarthria and ataxia in MS patients [61]. 

Cav2.2 is also upregulated in demyelinating areas [103], but its blocker pregabalin, prescribed to treat neuropathic pain in MS patients [62,63], might rise concerns about memory impairment, as it decreases LTP in mice and working memory in humans [63,64,65].

A mislocation of Kv is described in demyelinating areas of the MS brain [121]. Of note, the I_A_ K^+^ channel blocker Dalfampridine improves walking in MS patients and was approved by the FDA in 2010 [77]. 

Ion channels are also modulated by neurotransmitters, e.g., glutamate, GABA and purine receptors. Ligand-activated ionic channels, as AMPAR, NMDAR and P2X_7_R, lead to direct increase of intracellular Ca^2+^ that facilitates OPC migration [107,108,183,194,195,196], pointing to glutamate and ATP as chemoattractants for migrating OPCs. Glutamate ionotropic receptors generally lead to increased myelination, a concept that is supported by evidences indicating that firing axons are preferred targets for myelination. GABA, the main inhibitory neurotransmitter in the brain, generally improves myelination either by acting on GABA_A_Rs and GABA_B_Rs, even if contradictory results have been described on their role in OPC proliferation. Lastly, adenosinergic receptors are expressed on oligodendroglial cell lineage and operate a fine tuning of oligodendrogliogenesis (Figure 3). The Gs-coupled receptors, A_2A_R and A_2B_R, delay OPC differentiation by reducing K^+^ currents [117,118,170]. On the other hand, the Gi coupled A_1_R, as well as the purinergic-like GPR17 subtype, improves OPC differentiation and myelination [87,180,191,192,206]. 

The above data demonstrated the presence of a faint ionic channels/neurotransmitters receptors balance, whose interference is implicated in several neurodegenerative pathologies, including MS. The complex *scenario* of neuro-glial interaction, mediated by either or both ion channels and neurotransmitter receptors, should be considered to better understand the mechanisms leading to dysmyelination in CNS pathologies, and will be critical in evaluating the rationale for future MS treatments.

## Figures and Tables

**Figure 1 ijms-22-07277-f001:**
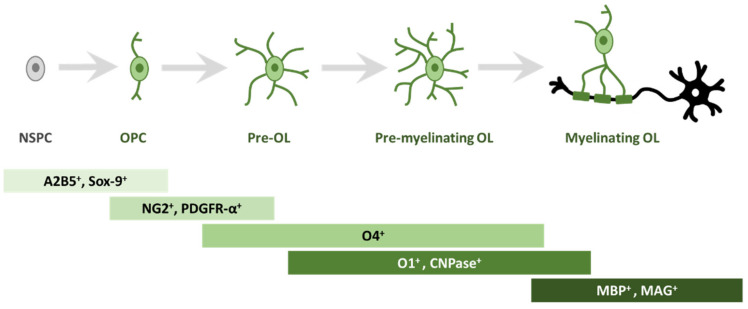
Schematic representation of morphological and antigen expression changes during oligodendrogliogenesis. A specific array of antigen expression and cell morphology has been associated to different steps of oligodendrogliogenesis, from oligodendrocyte precursor cells (OPC) to mature myelinating oligodendrocyte (OL). Abbreviations: cell surface ganglioside epitope (A2B5); SRY-Box Transcription Factor 9 (Sox9); neuron-glial antigen 2 (NG2) proteoglycan; receptor for PDGF-A (PDGFR-α); cell surface markers (O4); cell surface markers (O1); 2′, 3′-cyclic-nucleotide 3′-phosphodiesterase (CNPase); myelin basic protein (MBP); myelin associated glycoprotein (MAG).

**Figure 2 ijms-22-07277-f002:**
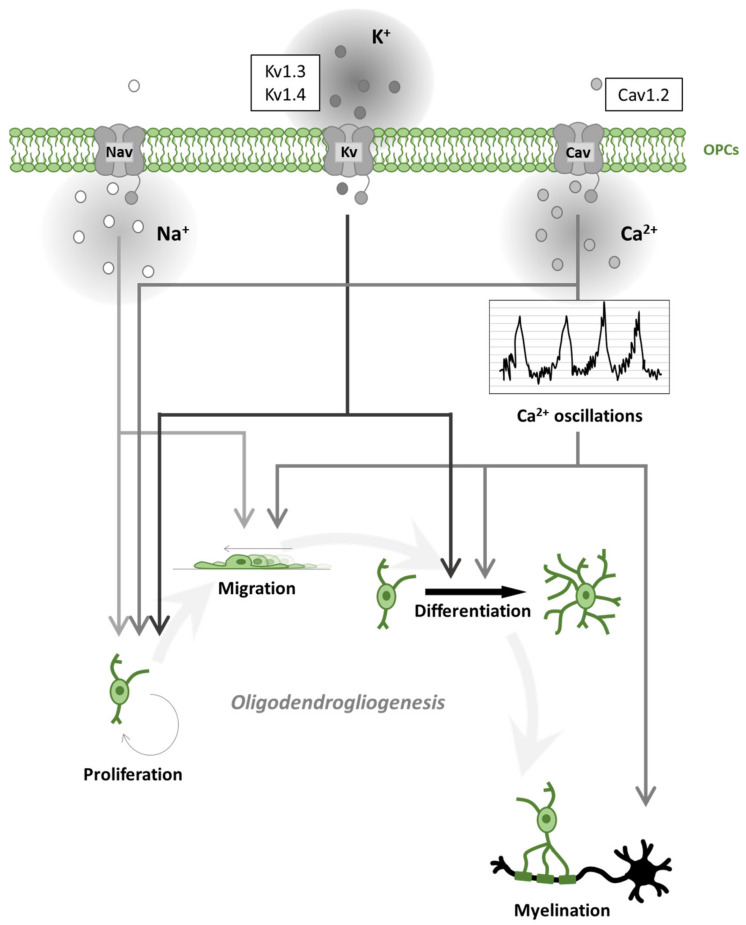
Oligodendrogliogenesis is differentially regulated by voltage-gated ion channels. Schematic representation of voltage-gated Na^+^, K^+^, Ca^2+^ channel (Nav, Kv, Cav) effects on oligodendrogliogenesis. Nav contributes to oligodendrocyte precursor cell (OPC) proliferation and migration. Kv regulates proliferation and differentiation. Cav regulates all steps of oligodendrogliogenesis. In particular, Cav activation contributes to spontaneous Ca^2+^ oscillations leading to accelerated migration, process formation (differentiation) and myelination. The most expressed ion channels, within each ion-selective group, are represented in the sidebar upwards.

**Figure 3 ijms-22-07277-f003:**
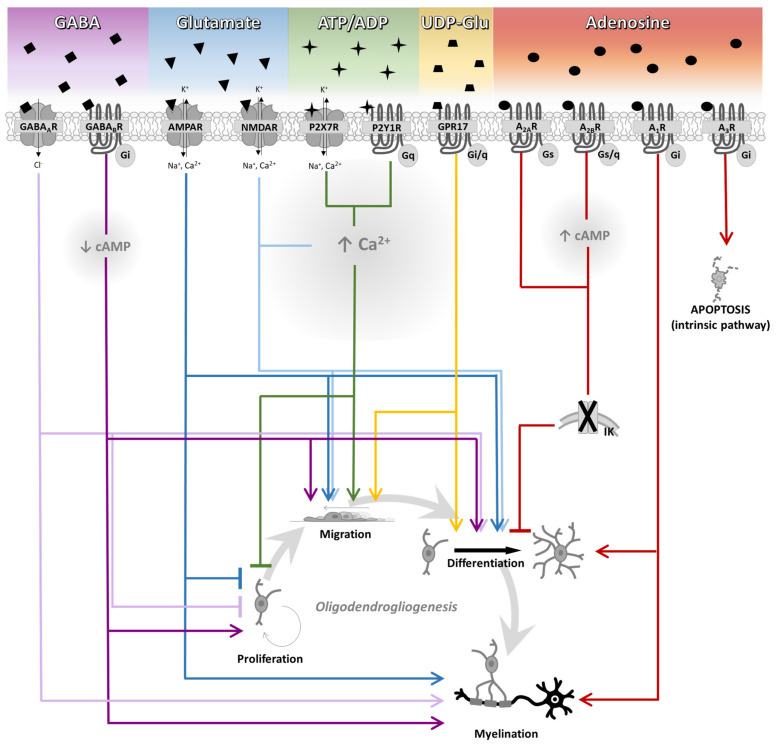
Regulation of oligodendrogliogenesis by neurotransmitters and neuromodulators. Schematic representation of the effect of different neurotransmitters and neuromodulators in oligodendrogliogenesis. γ-aminobutyric acid (GABA) promotes migration, differentiation, and myelination through both receptors. On the other hand, proliferation is enhanced by the metabotropic GABA_B_ receptor (GABA_B_R) stimulation, while it is inhibited by the ionotropic GAB_A_A receptor (GABA_A_R). Glutamate increases migration and differentiation by N-methyl-D-aspartate receptor (NMDAR) and α-amino-3-hydroxy-5-methyl-4-isoxazolepropionic acid receptor (AMPAR) activation. Moreover, NMDAR stimulation increases myelination and decreases proliferation. Both purine P2X and P2Y receptors (P2XR and P2YR) inhibit proliferation and lead to intracellular Ca^2+^ increase that promotes migration, as well as the Gi/q-coupled P2Y-like GPR17. The stimulation of Gs-coupled adenosine A_2A_ and/or A_2B_ receptors (A_2A_R and/or A_2B_R) lead to an increase in intracellular cAMP, which, in turn, closes IK channels and inhibits OPC differentiation. The activation of the Gi-coupled adenosine A_1_ and A_3_ receptors (A_1_R and A_3_R) induces myelination and apoptosis, respectively.

**Table 1 ijms-22-07277-t001:** Voltage-gated ion channel-targeting compounds and their role in demyelinating conditions.

Drug/s	Ion Channel/s	Preclinical/Clinical Trials	Effects
PF-01247324	Nav1.8-selective blocker	EAE	Improves motor coordination and cerebellar-like symptoms [53]
Safinamide	Unselective Nav blocker	EAE	Protects from neurological deficit and prevents microglial activation [54,55]
Flecainide	Nav blocker	EAE	Preserves axonal integrity and electrical conduction, reduces disability scores [56,57]
Phenytoin	Nav blocker	EAE Phase II	Preserves axonal integrity and electrical conduction, reduces disability scores [56,57]
			Neuroprotective in MS and related optic neuritis demyelination [58,59]
Lamotrigine	Nav blocker	EAE Phase II	Preserves axonal integrity and electrical conduction, reduces disability scores [56,57]
			Protective effects in preclinical models but no effect on cerebral volume changes [58,60]
			*Side effects*: reduction in white matter volume in secondary progressive MS patients [58,60]
Carbamazepine	Nav blocker	EAE Phase II	Improves paroximal dysarthria and ataxia in MS patients [61]
Pregabalin	Cav blocker	EAE Phase II	Reduces neuropathic pain in MS patients or EAE model.
			Neuroprotective during excitotoxicity or neuroinflammation in EAE.
			*Side effects*: reduces long-term potentiation in EAE mice and impairs memory function in MS patients [58,62,63,64,65]
Bepridil, Nitrendipine	L-type Cav1.x blocker	EAE	Reduces neuroinflammation and axonal pathology in EAE [66]
Nimodipine	L-type Cav1.2 blocker	EAE	Reduces EAE severity and demyelination [67]
			Antiapoptotic effect by preventing intracellular Ca^2+^ overload [68,69] Anti-inflammatory effect by preventing microglial activation [70,71,72,73]
BgK-F6A	Kv1.1 selective blocker	CPZ model EAE	Enhances remyelination in CPZ model [74]
			Reduces EAE severity [75]
Dalfampridine	I_A_ blocker	EAE FDA approved in 2010	Enhances axonal conduction in EAE [76]
			Improves motor activity (walking) in MS patients [77]
Glatiramer Acetate	Modulate K^+^, Cl^−^, Ca^2+^ and TRP channels [52]	FDA approved in 1996	Inhibits B lymphocytes maturation

Nav: voltage-gated sodium channels; Cav: voltage-gated calcium channels; Kv: voltage-gated potassium channels; I_A_: transient A-type potassium current; TRP: Transient Receptor Potential; EAE: Experimental Autoimmune Encephalomyelitis; CPZ: cuprizone; FDA: Food and Drug Administration; MS: Multiple Sclerosis.

**Table 2 ijms-22-07277-t002:** Expression and functional role of ion channels in oligodendroglial cells in vitro, in MS in vivo animal models or MS patients.

Ion Channel	OPC/OL Culture In Vitro		In Vivo MS Animal Models	MS Patients
Nav	Expression	TTX-sensitive Nav expressed in OPC, downregulated in OL [28,79,80,81,82,83,84,85,86,87,88]	Nav1.2 and Nav1.6 overexpressed in demyelinated sites (EAE) [53,90] Nav1.8 upregulated in cerebellar Purkinje cells (EAE) [91]	Nav1.8 upregulated in cerebellar Purkinje cells [91] Nav1.5 expressed in reactive astrocytes [92] Nav1.6 colocalizes with amyloid precursor protein in post-mortem tissues from secondary progressive MS brain [93]
	Function	↑ Migration and proliferation [94,95]		
Cav	Expression	L-type (Cav1.2, Cav1.3) in OPC (downregulated in OL) [81,83,96,97,98] N-type (Cav2.2) in OL [99]	Cav1.2 involved in remyelination (CPZ) [100,101] Increased activity of L-type in demyelinated corpus callosum (CPZ) [102] Overexpression of N-type (Cav2.2) in active lesions (EAE) [103]	Overexpression of N-type (Cav2.2) in active lesion [103]
	Function	L-type: ↑ Migration [104,105,106,107,108] ↑ Maturation [96,109]↑ Proliferation [86,96]		
Kir	Expression	Kir4.1 OPC > OL [98,106,110] Kir2.1, Kir7.1 and TASK1 in OL [111,112,113]	Kir4.1: ↑ Myelination [114,115]	Kir4.1: High level in serum from MS patients [116]
	Function	↑ Maturation [115]		
Kv	Expression	I_A_ and I_K_ OPC > OL [79,80,81,82,117,118] Kv1.2-6, Kv2.1, Kv7.2 in OPC [110,119,120] Kv4.2, Kv4.3, and Kv3.3. in OPC KCa1.1 in OPC [110]	Kv mislocalization in EAE animals [121] Kv1.2, Kv1.4, and Kv2.1 downregulated in EAE [122]	Kv mislocalization in post-mortem human MS lesions [121]
	Function	↑ Maturation [87,123] Kv1.3, Kv1.4: ↑ proliferation Kv1.6: ↓ proliferation		

Nav: voltage-gated sodium channels; Cav: voltage-gated calcium channels; Kir: inward-rectifier potassium currents; Kv: voltage-gated potassium channels; I_A_: transient A-type potassium current; I_K_: delayed-rectifier potassium currents; KCa: calcium-activated potassium channels; EAE: Experimental Autoimmune Encephalomyelitis; CPZ: cuprizone; MS: Multiple Sclerosis. ↑: increase of. ↓: decrease of.

**Table 3 ijms-22-07277-t003:** Expression and functional role of ligand-gated ion channels in oligodendroglial cells in vitro and in MS in vivo animal models or MS patients.

Ligand	Receptor	In Vitro OPC Cultures	In Vivo MS Animal Models/MS Paitens
Glutamate	AMPAR	↓ proliferation and ↑ maturation [109,173,174,175]	↑ myelination [176,177]
		↑ migration [107]	
	NMDAR	↑ migration and ↑ differentiation [178]	?
GABA	GABA_A_R	↓ myelination [179]	↑ OPC proliferation
			↓ OPC differentiation [165]
			↑ myelination [165,180]
	GABA_B_R	↑ proliferation and ↑ migration [181]	?
		↑ differentiation [182]	
ATP/ADP	P2X_7_R	↓ proliferation and ↑ migration [183]	OL damage and myelin loss during ischemia or neuroinflammation [184,185]
			EAE-induced OL death by Ca^2+^ overloading [186]
	P2Y_1_R	↑ migration [187,188]	?
		↓ proliferation [188]	
	P2Y_12_R	?	Downregulated in the cerebral cortex of post-mortem MS brains [113,189]
Uracil-nucleotides	GPR17R	↑ migration [87]	Overexpressed in active lesion of post-mortem MS brains [190]
		↑ differentiation [87,191]	
Adenosine	A_2A_R/A_2B_R	↓ differentiation [117,118]	?
	A_1_R	↓ proliferation [192]	↑ myelination [170]
		↑ differentiation [192]	
	A_3_R	Induces OL apoptosis [170]	?

OPC: oligodendrocyte precursor cell; OL: oligodendrocyte; AMPAR: α-Ammino-3-idrossi-5-Metil-4-isossazol-Propionic Acid receptor; NMDAR: N-methyl-D-aspartate receptor; GABA: γ-aminobutyric acid; GABA_A_R: GABA_A_ receptor; GABA_B_R: GABA_B_ receptor; ATP: Adenosine triphosphate; ADP: Adenosine diphosphate; P2X_7_R: P2X_7_ purinergic receptor; P2Y_1_R: P2Y_1_ purinergic receptor; P2Y_12_R: P2Y_12_ purinergic receptor; GPR17R: P2Y-like receptor; A_2A_R: A_2A_ adenosine receptor; A_2B_R: A_2B_ adenosine receptor; A_1_R: A_1_ adenosine receptor; A_3_R: A_3_ adenosine receptor; EAE: Experimental Autoimmune Encephalomyelitis; MS: Multiple Sclerosis. ↑: increase of. ↓: decrease of. ?: unknown.

## Data Availability

Not applicable.

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
