# Peer review of "Ion Channels as New Attractive Targets to Improve Re-Myelination Processes in the Brain"

_ijms, 2021, doi:10.3390/ijms22147277_

Round 1

Reviewer 1 Report

This is an interesting and well written review. My only concern regards the title. In fact, the manuscript includes informations not only on the role of voltage-gated ion channels in oligodendrocytes and during demyelination, but also on the role of neurotransmitters (GABA, glutamate and purines) and related ligand-gated ion channels.

A table summarizing the pharmacological effects of ion channels modulators described troughout the text could be helpful to the reader.

Some minor points and typos need correction:

The subtitle “Nav (or Cav, Kv) in demyelination” should be uniformed as “Nav channels after demyelination”  or “Nav channels in demyelinating disease”,

In the abstract:

-Multiple sclerosis (MS) is the most common demyelinating disease of the central nervous system (CNS) characterized by neuroinflammation and neurodegeneration”.

-please, correct: under demyelinating con-ditions, re-myelinate, com-munication , sum-marizes, physio-pathophysiological 

-the contribution of NCX to calcium oscillations in differentiating oligodendrocytes need to be considered and more papers should be cited.

Author Response

Q1: This is an interesting and well written review. My only concern regards the title. In fact, the manuscript includes informations not only on the role of voltage-gated ion channels in oligodendrocytes and during demyelination, but also on the role of neurotransmitters (GABA, glutamate and purines) and related ligand-gated ion channels.

R1: We thank the Reviewer for his/her positive evaluation of our manuscript and for the constructive criticisms which significantly improved our work. As suggested, we changed the title from:

Voltage-gated channels: new attractive targets to improve re-myelination processes in the brain”

to:

Ion channels as new attractive targets to improve re-myelination processes in the brain

Q2: A table summarizing the pharmacological effects of ion channels modulators described troughout the text could be helpful to the reader.

R2: As suggested, we added a table (Table 1 in the revised version of the manuscript, page 6) to summarize the pharmacological effects of ion channels modulators tested in demyelinating pathologies and described throughout the text.

Some minor points and typos need correction:

Q3: The subtitle “Nav (or Cav, Kv) in demyelination” should be uniformed as “Nav channels after demyelination”  or “Nav channels in demyelinating disease”

R3: The subtitles “Nav (or Cav) in demyelination” have been changed to:

3.1.1. Nav channels in demyelinating diseases” (page 8 line 216)

3.2.1. Cav channels in demyelinating diseases” (page 10 line 339).

The subtitle regarding K+ channels has been uniformed as follows:

3.3.1. Voltage-gated K+ channels in demyelinating diseases” (page 12 line 449) because it does not relate to Kv only but also to Kir channels.

Q4 -In the abstract: Multiple sclerosis (MS) is the most common demyelinating disease of the central nervous system (CNS) characterized by neuroinflammation and neurodegeneration”.

R4 – The sentence has been changed as suggested (Abstract, page 1 line 9).

Q5 -please, correct: under demyelinating con-ditions, re-myelinate, com-munication , sum-marizes, physio-pathophysiological 

R5 – We thank the Reviewer for his/her kind revision. However, above mentioned spelling issues are made automatically by the formatting template of the Journal. We will make sure to have the correct form in the proofreading phase.

Q6 -the contribution of NCX to calcium oscillations in differentiating oligodendrocytes need to be considered and more papers should be cited.

R6 – We thank the Reviewer for raising this important point. We added a paragraph (page 8 lines 233-239) to describe the role of NCX in oligodendrocyte differentiation and myelination, as follows:

Of note, data demonstrated that NCX3 is strongly upregulated during oligodendrocyte development (Boscia et al., Cell Death Differ. 2012 Apr;19(4):562-72) and its activity is required for the maintenance of Ca2+ transients necessary to myelin synthesis in oligodendrocyte cultures (Boscia et al., cell Calcium 2020 Jan;85:102130). Recent evidence confirmed this hypothesis and deepened knowledge in the field by demonstrating that neuronal activity, presumably in form of local extracellular [K+] changes, might locally influence NCX-mediated Ca2+ transients in OLs thus triggering local MBP synthesis in the vicinity of an active axon (Hammann et al., Cell Calcium. 2018 Jul;73:1-10).”

Reviewer 2 Report

Cherchi and colleagues present a paper entitled" Voltage-gated channels: new attractive targets to improve re-myelination processes in the brain."

The authors detailed recent discoveries that consider ion channels, neurotransmitters/neuromodulator receptors that regulate oligodendrocyte differentiation, favouring communication between glial cells and neurons. Overall, the review is exciting and well organized.

Considering that there is much information, to facilitate the reader to have a quick overview, it might perhaps be helpful to insert one or more tables where the main functions are highlighted for the various oligodendrocyte markers, ion channels and neurotransmitters, well detailed in the text, both in normal OL differentiation and in demyelinating diseases.

Author Response

Cherchi and colleagues present a paper entitled" Voltage-gated channels: new attractive targets to improve re-myelination processes in the brain."

The authors detailed recent discoveries that consider ion channels, neurotransmitters/neuromodulator receptors that regulate oligodendrocyte differentiation, favouring communication between glial cells and neurons. Overall, the review is exciting and well organized.

Q1. Considering that there is much information, to facilitate the reader to have a quick overview, it might perhaps be helpful to insert one or more tables where the main functions are highlighted for the various oligodendrocyte markers, ion channels and neurotransmitters, well detailed in the text, both in normal OL differentiation and in demyelinating diseases.

R1. We thank the Reviewer for his/her positive evaluation of our manuscript and for the constructive criticisms which significantly improved our work. As suggested, we added new tables to facilitate the reader to have a quick overview of ion channels/neurotransmitters involved in demyelination-remyelination processes and various compounds tested in different in vitro or in vivo paradigms, as follows:

"Table 1: Ion channel-targeting compounds and their role in demyelinating conditions."

"Table 2: Expression and functional role of ion channels in oligodendroglial cells in vitro, in MS in vivo animal models or MS patients."

"Table 3: Expression and functional role of ligand-gated ion channels in oligodendroglial cells in vitro and in MS in vivo animal models or MS patients."
